# Sleep Quality as a Predictor of Quality-of-Life and Emotional Status Impairment in Patients with Chronic Spontaneous Urticaria: A Cross-Sectional Study

**DOI:** 10.3390/ijerph20043508

**Published:** 2023-02-16

**Authors:** Manuel Sánchez-Díaz, Juan Ángel Rodríguez-Pozo, José María Latorre-Fuentes, Maria Carmen Salazar-Nievas, Molina-Leyva Alejandro, Salvador Arias-Santiago

**Affiliations:** 1Dermatology Unit, Hospital Universitario Virgen de las Nieves, 18014 Granada, Spain; 2Biosanitary Institute of Granada (ibs.GRANADA), 18002 Granada, Spain; 3Urticaria Clinic, Hospital Universitario Virgen de las Nieves, 18014 Granada, Spain; 4Dermatology Department, School of Medicine, University of Granada, 18071 Granada, Spain

**Keywords:** urticaria, sleep quality, quality of life, mood status disturbances

## Abstract

Chronic Spontaneous Urticaria (CSU) leads to a decreased quality of life in patients because of pruritus and skin lesions. However, there is still little evidence on the impact that a worse sleep quality could have on the quality of life and emotional disorders in these patients. The aims of the present study are to analyze the potential impact of sleep quality on the quality-of-life and emotional status of patients with CSU. A cross-sectional study of 75 CSU patients was performed. Socio-demographic variables and disease activity, quality of life, sleep, sexual disfunction, anxiety, depression and personality traits were collected. A majority of 59 of the patients suffered from poor sleep quality. Sleep quality impairment was associated with worse disease control, greater pruritus and swelling and poorer general and urticaria-related quality-of-life (*p* < 0.05). Patients with poor sleep quality showed an increased prevalence of anxiety (1.62-fold) and depression risk (3.93-fold). Female sexual dysfunction, but not male, was found to be linked to poorer sleep quality (*p* = 0.04). To conclude, sleep quality impairment in patients with CSU is related to poor quality-of-life, worse disease control and higher rates of anxiety and depression. Global management of the disease should take sleep quality into account to improve the care of CSU patients.

## 1. Introduction

Chronic spontaneous urticaria (CSU) is characterized by the appearance of typical urticarial lesions for more than six weeks [1]. This disorder produces pruritus of variable intensity that affects the quality of life of patients, without any underlying triggering factor and appearing most days [2]. Although CSU usually disappears spontaneously with the course of time, generally within a period of 5 years, some cases show a longer duration [3,4]. Most commonly used treatments include the use of high-dose antihistamines, with omalizumab being the treatment of choice for cases resistant to antihistamines [2,5]. CSU has been described to impact negatively on the quality of life of patients as a result of the pruritus and visible skin lesions [6,7], therefore leading to emotional status disturbances and poor quality of sleep [8,9].

Currently, there are a considerable number of studies that explore the impact of poorer sleep quality on the quality-of-life of patients suffering from other pruritic conditions, such as psoriasis [10] or atopic dermatitis [11]. However, evidence of the impact of sleep quality on patients with urticaria has been very scarcely discussed in the scientific literature [9], with scientific reports showing a poor quality of sleep and poor quality of life, but without exploring the impact on emotional status and other spheres of the quality of life. More in-depth research on these topics would be of great interest because of the increasing evidence on the role that poor sleep quality could play in the development of significant neuro-psychiatric [12,13] and cardiovascular [14] complications.

Given that the potential connection of sleep quality, poorer quality of life and disturbed emotional state has not yet been clearly studied, the objectives of this work are to analyze the effect that sleep quality could have on various dimensions of quality-of-life and personality traits of patients with CSU, as well as to investigate the association of sleep quality with emotional disturbances in patients with CSU. 

## 2. Materials and Methods

Design: A cross-sectional study was performed. Patients with diagnosis of CSU, with independence of the severity, were included in the study with the objective of exploring the impact of sleep quality on quality of life and emotional status disturbances. The STROBE checklist was followed when performing the study.

Patients: Patients were recruited from the specialized Urticaria Clinic of the Virgen de las Nieves University Hospital and from “Asociación de Afectados por Urticaria Crónica”, the official Spanish patient association for patients with CSU. Those patients attending the Urticaria Clinic were invited to fill out an online questionnaire after their protocolized follow-up consultation. The inclusion period ranged from January 2020 to August 2021. 

Inclusion criteria: Patients included in the study had to meet the following criteria: (a) They had to be diagnosed as having CSU, with independence of the severity and the treatment; (b) They had to be aged 18 years old or older; (c) They had to give their informed consent to participate in the study. 

Exclusion criteria: The exclusion criteria considered were as follows: (a) Patients declining to participate in the study; and (b) Patients suffering from major diseases which could negatively impair their quality of life.

### 2.1. Variables of Interest

#### 2.1.1. Main Variables

Variables with respect to disease severity and those related to quality-of-life assessment were included as main variables. On the one hand, disease severity-related variables were as follows: Urticaria Control Test (UCT) was considered to explore disease control. It consists of a Likert scale composed of 8 different questions on physical symptoms and quality-of-life in the last 4 weeks. Final score varies from 0 (meaning lack of control) to 32 (meaning total control) [15].Other variables were the age of onset of CSU, the disease duration, date and the current treatments.

On the other hand, variables exploring quality-of-life, emotional status disturbances, sleep quality and sexual impairment were collected using validated questionnaires:The Pittsburgh Sleep Quality Index (PSQI) Questionnaire was used to explore sleep quality. PSQI values ranges from 0 (best sleep quality) to 21 (worst sleep quality) points. A cut-off point of scores higher than 5 was considered significant as sleep quality impairment [16].Dermatology Life Quality Index (DLQI) was employed as the indicator of general quality of life. It is composed of 10 items with values from 0 to 3. Global score varies from 0 (being the least affected) to 30 (the most affected) [17].Chronic Urticaria Quality of Life Questionnaire (CUQoL) was used as CSU-related quality of life marker. It includes physical, emotional and social issues related to CSU. Twenty-three Likert-type items are scored from 1 (never) to 5 (very much), finally obtaining a range of 0 (no quality-of-life impairment) to 100 (maximum quality-of-life impairment). Subscales for pruritus, swelling, daily activities, sleep impairment, daily limitations and physical appearance are included in the questionnaire, as well as the overall CUQoL score [18].The Hospital Anxiety and Depression Scale (HADS) was used to explore emotional status disturbances. It consists of a Likert scale which is composed of two parts: anxiety evaluation (odd-numbered questions) and depression evaluation (even-numbered questions). A cut-off point of ≥8 on any of the subscales was considered representative of anxiety or depression, respectively [19].The DS14 Questionnaire was employed to assess the traits of Type D Personality (TDp). It is composed of 14 questions for both components of TDp: negative affectivity (7 questions) and social inhibition (7 questions). A cut-off point of ≥10 in the two components is established as an indicator of TDp [20,21].The International Index of Erectile Function (IIEF-5) [22] and Female sexual function Index (FSFI-6) [23] questionnaires were used to explore sexual function impairment in men and women respectively. IIEF-5 assesses sexual function in males with scores ≤ 21 being considered significant. FSFI-6 evaluates female sexual function and a score ≤ 19 was considered as indicative of dysfunction.Numeric Rating Scale (NRS) for sexual function impairment was recorded: this ranged from 0 to 10 and measured the degree of sexual dysfunction associated with the CSU [24].

#### 2.1.2. Other Variables of Interest

Data on clinical variables and socio-demographic variables were collected. These included sex, age, history of previous diseases, treatment history for CSU, marital status and educational level.

### 2.2. Statistical Analysis

Descriptive statistics were used to analyze the sample characteristics. The Shapiro-Wilk test was applied to determine the normality of the data. Mean and standard deviation (SD) were used to report continuous data. Relative and absolute frequency distributions were used for qualitative data. The χ^2^ test or Fisher's exact test were employed to make comparisons between nominal variables and Student's t test or the Wilcoxon-Mann-Whitney test were used to perform comparisons between nominal and continuous data. To further investigate potential related factors, associations between continuous data were evaluated by simple linear regression. The β coefficient and SE (standard error) were considered as predictors of the dependent variable values. p values less than 0.05 were considered as establishing statistical significance. Statistical analyses were performed with JMP version 14.1.0 (SAS Institute, Cary, NC, USA). 

## 3. Results

### 3.1. General Characteristics of the Patients

Seventy-seven CSU patients were invited to participate in the study, with most of them (75/77—97.4%) completing the questionnaires. A majority was contacted through the CSU association (51/75—68%). No differences were found between patients collected by each source, neither with respect to sex, age, disease activity, evolution time, occupation, nor in regard to marital status (*p* > 0.60). 

Socio-demographic characteristics can be seen in Table 1. As a brief summary, mean age was 46.5 (SD 11.2) years old, with a female-to-male ratio of 2.40. A majority of the patients was actively working (59/75—78.67%), and were mostly in a relationship (58/75—77.3%).

### 3.2. Clinical and Quality-of-Life Characteristics of the Patients

Disease characteristics can be seen in Table 1. Relevant data include a long-lasting mean duration of the disease (10.7 years, SD 10.7) and the presence of patients with a variety of treatments, including omalizumab (29/75—38.67%), antihistamines (31/75—41.33%) and some with no current treatment (8/75—10.66%). 

On the other hand, scores for the quality-of-life questionnaires can be seen in Table 1. There was a high prevalence of emotional disorders including anxiety (35/75—46.7%) and depression (31/75—41.33%). An impaired sexual function was found in 58.66% of the patients (44/75), including females (30/55—54.54%) and males (14/22—63.63%). 

### 3.3. Association of Sleep Quality with Socio-Demographic and Characteristics in Patients with CSU

Mean PSQI score in the sample was 9.53 (SD 4.68). When establishing the cut-off point for sleep quality impairment, it was found that a majority of 78.67% (59/75) of the patients suffered from poor quality of sleep. After univariate analysis (Table 2), no differences were found in terms of socio-demographic characteristics between patients with impaired quality of sleep and patients with normal quality of sleep, except for a higher rate of unemployed patients having sleep quality disorders (*p* = 0.06). However, it was found that patients with poor sleep quality had worse disease control when compared to those with normal sleep quality (*p* = 0.01).

### 3.4. Association of Sleep Quality and CSU Disease Control, Symptoms and Signs:

A worse disease control was found in patients with sleep quality impairment (*p* = 0.01, Table 1, Figure 1). Moreover, CSU symptoms and signs were associated with poorer quality of sleep (Table 3). In this regard, patients with impaired sleep quality showed higher scores in the pruritus DLQI subscale (*p* = 0.07), as well as worse CUQOL pruritus and swelling subscales (*p* < 0.05).

### 3.5. Association of Sleep Quality and Quality-of-Life Scores, Emotional Status Scores and Sexual Function in Patients with CSU

A worse quality of life was found for those patients having impaired sleep quality (Table 3, Figure 1). This can be seen in higher overall DLQI scores (*p* = 0.02), as well as in DLQI subscales interference in daily activities, social relationships (*p* < 0.05) and treatment inconveniences (*p* = 0.07). CUQOL score showed similar results, with patients with sleep quality impairment having worse overall CUQOL scores (*p* = 0.01), as well as on CUQOL subscales for sleep disturbances and physical aspect (*p* < 0.05). 

Regarding emotional status disturbances, higher scores for anxiety and depression were found in patients with impaired quality of sleep (*p* < 0.01) (Table 3, Figure 1). Actually, it was found that having impaired sleep quality increased the prevalence of anxiety by 1.62-fold (Odds Ratio 1.62–CI (1.03–3.51)) and multiplied depression risk 3.93-fold (Odds Ratio 3.93–CI (1.04–14.75).

Finally, lower sexuality scores were found in woman with impaired sleep quality when compared to those with normal sleep quality (*p* = 0.04), with no differences for males, nor in terms of NRS for sexual activity.

## 4. Discussion

As shown by the results of the present study, sleep quality impairment in patients with CSU seems to be associated with a variety of worse quality-of-life indexes, as well as with worse disease control and higher rates of anxiety and depression. Moreover, it seems to have an elevated prevalence in these patients and seems to be related to disease signs and symptoms, such as pruritus and swelling. 

To date, few reports have analyze the impact of sleep quality impairment in patients with CSU. The main study found in relation to this issue analyzed a sample of 21 CSU patients and a sample of controls [9]. In this report, patients with CSU were more likely to suffer from sleepiness and apnea-hypopnea symptoms. Moreover, in patients with CSU, worse quality of life was found to be related to greater sleep latency. Other studies show similar results [25]. These results are in line with those found in our study: CSU is linked to bad sleep quality in a remarkable percentage of the patients, and sleep quality impairment is related to poorer quality of life. On the other hand, no studies have properly explored the issue of sleep quality, emotional status disturbances and sexual function in these patients to date. 

In light of these results, it could be hypothesized that patients with CSU could have higher rates of anxiety and depression which would lead to impaired sleep quality or, on the contrary, that those patients with CSU and worse sleep quality would be more likely to develop mood status disturbances. Regardless of the causal order of these associations, it seems clear that the relationship between poor sleep quality and mood disorders is homogeneous in chronic skin conditions, such as atopic dermatitis [11], psoriasis [10] and alopecia areata [26].

On the other hand, several studies show how sleep disorders can lead to higher levels of different inflammatory and autoimmune markers in humans [27,28], such as higher levels of C-Reactive Protein, Interleucin-6 or Tumor Necrosis Factor. Furthermore and conversely, inflammatory markers can also be responsible for sleep disorders in preclinical models [27]. Study of whether the observed relationship between CSU, sleep disturbances and their impact on quality of life is mediated by alterations in markers of inflammation would be of great future interest, in order to understand the pathophysiology underlying the associations found. 

Finally, since sleep quality impairment seems to be associated with poor disease control, pruritus and swelling, proper treatment of CSU patients should be essential in order to offer our patients global and quality management of their disease.

The main limitations of the present study include the cross-sectional design which could limit the causal inferences, the limited sample size and the lack of control group, which makes it impossible to compare the prevalence of sleep disorders with healthy population, therefore only allowing exploration of the associations of sleep quality impairment in patients with CSU. 

## 5. Conclusions

Sleep quality impairment is frequent among patients suffering from CSU and seems to be associated with poorer quality of life, greater disease symptoms and signs and a higher rate of anxiety ang depression. Global management of the disease should take sleep quality into account to improve the care of CSU patients. 

## Figures and Tables

**Figure 1 ijerph-20-03508-f001:**
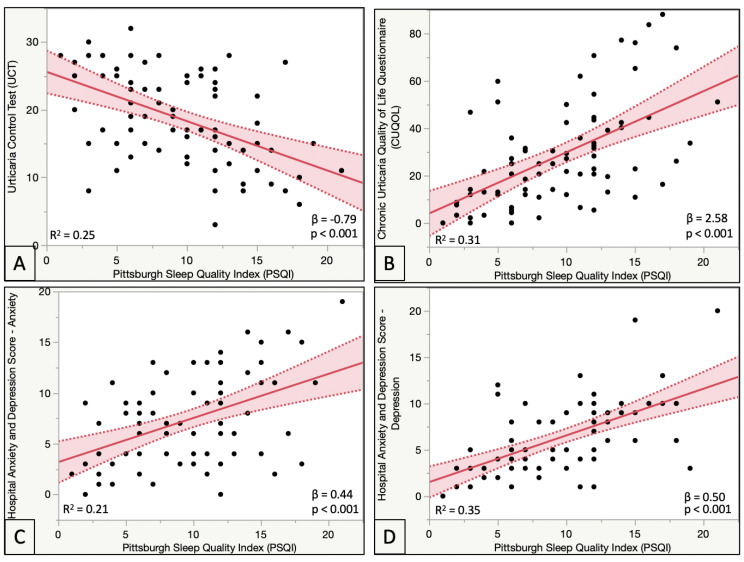
Correlations between PSQI and disease control (UCT), quality of life (CUQOL), Anxiety (HADS-Anxiety) and Depression (HADS-Depression). Worse quality of life (higher PSQI scores) is related to worse disease control (**A**), worse quality of life (**B**), higher anxiety scores (**C**) and higher depression scores (**D**).

**Table 1 ijerph-20-03508-t001:** Sociodemographic features of the sample and characteristics of the disease.

Variables All Patients (*n* = 75)					
Socio-demographic features					
Age (years)	46.48 (SD 11.25)		Marital status	With couple	77.3% (58/75)
				Single	22.7% (17/75)
Sex (%)	Male:	29.3% (22/75)	Occupation	Employed (active job)	78.7% (59/75)
	Female:	70.7% (53/75)		Unemployed (without active job)	21.3% (16/75)
Educational level	Basic education		8% (6/75)	Professional education	28% (21/75)
	Secondary education		13.3% (10/75)	University education	50.7% (38/75)
Disease characteristics					
Disease duration (years)	10.73 (SD 10.69)		Urticaria Control Test score	18.64 (SD 6.90)	
Disease duration	<10 years	60% (45/75)	Current treatment for CSU	Antihistamines	41.3% (31/75)
				Antihistamines + corticosteroids	6.7% (5/75)
				Corticosteroids	2.7% (2/75)
	>10 years	40% (30/75)		Omalizumab	38.7% (29/75)
				No medical treatment	10.6% (8/75)
Quality of life indicators					
DLQI	8.12 (SD7.15)		Overall CUQOL	28.63 (SD 21.73)	
DS14 (% of positive test)	28% (21/75)		PSQI	9.53 (SD 4.68)	
HADS Depression (% of positive test)	41.33% (31/75)		HADS Anxiety (% of positive test)	46.67% (35/75)	
FSFI (% of female sexual dysfunction)	54.54% (30/55)		IIEF (% of male sexual dysfunction)	63.63% (14/22)	

**Table 2 ijerph-20-03508-t002:** Association of sleep quality with sociodemographic characteristics and disease control in CSU patients.

	Impaired Sleep Quality (PSQI > 5) (*n* = 59) Mean/% (SD, Fraction)	Normal Sleep Quality (PSQI < 5) (*n* = 16) Mean/% (SD, Fraction)	*p* Value
Age (years)	45.79 (SD 1.46)	49 (SD 2.81)	0.31
Sex (female)	72.88% (43/59)	62.50% (10/16)	0.42
Marital Status (couple)	25.86%/35.27%	74.14%/64.70%	0.45
Occupation (Employed)	74.58% (44/59)	93.75% (15/16)	0.06
Duration of the disease (years)	9.74 (SD 1.37)	14.37 (SD 2.64)	0.12
Current treatment for CSU (Omalizumab)	38.9% (36/59)	37.50% (10/16)	0.91
Urticaria Control Test	17.67 (SD 0.87)	22.25 (SD 1.67)	0.01

**Table 3 ijerph-20-03508-t003:** Association of sleep quality impairment with quality-of-life indexes, emotional status disturbances and sexual function in patients with CSU.

	Impaired Sleep Quality (PSQI > 5) (*n* = 59) Mean (SD)	Normal Sleep Quality (PSQI < 5) (*n* = 16) Mean (SD)	*p* Value
Overall DLQI	9.05 (SD 0.90)	4.69 (SD 1.74)	0.02
DLQI—Pruritus, pain, discomfort	1.50 (SD 0.12)	1.00 (SD 0.24)	0.07
DLQI—Embarrassment, self-consciousness	1.08 (SD 0.13)	0.62 (SD 0.26)	0.13
DLQI—Interference in daily activities	0.77 (SD 0.10)	0.18 (SD 0.20)	0.01
DLQI—Chose of clothes	1.32 (SD 0.14)	0.81(SD 0.27)	0.10
DLQI—Social or leisure activities	0.93 (SD 0.12)	0.50(SD 0.24)	0.12
DLQI—Sport	0.91 (SD 0.13)	0.50 (SD 0.25)	0.15
DLQI—Work and study	0.64 (SD 0.11)	0.37 (SD 0.21)	0.27
DLQI—Social relationships	0.67 (SD 0.10)	0.25 (SD 0.21)	0.03
DLQI—Sexual difficulties	0.77 (SD 0.13)	0.37 (SD 0.26)	0.17
DLQI—Treatment inconveniences	0.40 (SD 0.08)	0.06 (SD 0.16)	0.07
Overall CUQOL	31.85 (SD 2.72)	16.77 (SD 5.24)	0.01
CUQOL—Pruritus	42.37 (SD 3.82)	28.12 (SD 7.24)	0.04
CUQOL—Swelling	19.91 (SD 3.03)	6.25 (SD 5.83)	0.04
CUQOL—Daily activities	26.20 (SD 3.14)	15.10 (SD 6.03)	0.11
CUQOL—Sleep disturbances	38.30 (SD 3.13)	13.75 (SD 6.02)	0.00001
CUQOL—Limitations	31.07 (SD 3.43)	25 (SD 6.59)	0.41
CUQOL—Aspect	33.22 (SD 3.33)	16.56 (SD 6.39)	0.02
HADS—Anxiety score	8.03 (SD 3.54)	5.75 (SD 1.04)	0.007
HADS—Depression score	7.10 (SD 0.48)	3.43 (SD 0.92)	0.0008
NRS sexual activity	3.66 (SD 0.48)	3.43 (SD 0.92)	0.83
FSFI index	14.79 (SD 1.27)	19.90 (SD 2.64)	0.04
IIEF index	19 (SD 1.08)	19.83 (SD 1.376	0.80

## Data Availability

The data presented in this study are available on reasonable request from the corresponding author.

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
