# Peer review of "Sleep Quality as a Predictor of Quality-of-Life and Emotional Status Impairment in Patients with Chronic Spontaneous Urticaria: A Cross-Sectional Study"

_ijerph, 2023, doi:10.3390/ijerph20043508_

Round 1

Reviewer 1 Report

Review

Introduction:

The introduction provides relevant background information and clearly states the research gap to be filled.

Materials and Methods:

The authors state that they used simple linear regression (i.e., ordinary least squares); however, they also state that they used beta coefficients to predict the log odds of the dependent variable. This is confusing because simple linear regression does not require logistic regression transformations. This suggests an error in analysis. On line 130 the authors state that they used beta coefficient and “SD” as predictors of the log odds ratio. Does this mean “SD” mean standard deviation? I am assuming that the authors mean to say “SE” for standard error. All of which is confusing because there is no mention of logistic regression.

For the main variables, there is no mention of validity or reliability for any of the scales the authors used. I suggest some tests of internal consistency and reporting Cronbach’s alpha for reliability.

It is not precisely clear which variables were treated continuously or categorically.

Overall, there is a lot to be desired in the methods section. However, simple explanations could suffice. Were the regression tests adjusted for sociodemographic characteristics? What variables were entered into each regression? How did the authors decide which variables to include during model development?

Results:

There is language that is interpretive in the results section. For example, line 137 states the word “finally” completing the questionnaires and line 146 states “the most relevant data…”. These are two examples of language that should not be included in the results section.

Table 3 needs more descriptive headings. Are the values means? Odds ratios? Betas? What test is being used? My suggestion is to add a note that describes the tests.

Discussion:

The discussion does well to explain how the current set of findings fits in with broader research. There is no discussion of potential confounding or other explanatory pathways for the phenomenon observed. The authors state that the directionality of causality cannot be determined, which is fine given the research design. However, there is no discussion of what else could be causing the association between CSU and, say, sleep disturbance. Are there no alternative explanations for the phenomenon described?

Author Response

Dear Reviewer,

The authors of the manuscript would like to thank you for your comments, as they allow us to improve the scientific quality of our research. Below you can see a point-by-point response to your comments: 

  • As you suggested, there was a mistake in the "Statistics" section. Simple linear regression was performed, and therefore no "log odds" were present. No logistic regression was performed therefore.
  • No multivariate models were performed, as only univariate analysis are shown. All the variables regarding DLQI and CUQOL were treated as continuous.
  • Interpretative language has been removed from results section to make it more objective.
  • Table 3 has been improved.
  • We have highlighted the inflammatory markers as a potential causal mediation between CSU and sleep disorders. This could represent an alternative explanation for the phenomenon which is described.

Reviewer 2 Report

A very interesting cross-digital study on 75 patients affected by chronic spontaneous urticaria, with almost 80% of them suffering from poor sleep quality. These patients were linked with a variety of worse quality-of-life indexes, as well as with worse disease control and higher rates of anxiety and depression.

Although the number of patients included in the study is not high, I think that this paper will be eligible to be published after minor revisions:

Line 32-34 "Although CSU usually disappears spontaneously with the course of time, generally within a period of 5 years, some cases show a longer duration." this sentence needs some references, such as: doi: 10.3390/pharmaceutics14020294. and doi: 10.1111/dth.15111

Author Response

Dear reviewer, 

Thank you very much for your comments. The suggested references have been included in the text. 

Reviewer 3 Report

The authors present a small but very interesting and imprtant moncenter study on the impact of chronic urticaria on sleep quality.

While the study is generally well-conducted and the manuscript reads sound, some points are suggested to be addressed and considered by the authors:

-          The reference to CU-QoL (9) seems not to refer to the right publication. Please check and include accordingly (line 93 – 99)

-          In line 52 is written “…patients with AA” – I guess it should be “CSU” instead

-          “Emotional status disorders” which is repeatedly used, is not a commonly and widely used term. Suggest to use the term “emotional disorder” instead.

-          In line 18 “A majority of 78.67% of the patients…” – better to mention the absolute number instead of percentage.

-          For the UCT the long Spanish version was used – what is the cut-off in this version? Internationally, the validated short version of the UCT with a maximum score of 16 points and a suggested cutoff of 12 points is usually used. In addition, the long Spanish version of the UCT includes quality of life questions that are also covered by the CU-QoL. It is suggested to also make a calculation on the short UCT version, which is part of the long version and present these numbers when referring to disease control. A recalculation and adoption of the values in Figure 1 and the corresponding Tables is recommended.

-          There are some language related mistakes, proofreading by a native speaker is should be considered.

-          With regard to the therapeutic strategy in CSU, it is recommended to use the algorithm from the 2020 international guideline and to attach it as a reference.

-          Why do 10% of the patients not receive treatment for their CSU? How does this correlate to their UCT and QoL data?

-          What does “protocolized follow-up” of patients mean? Had all patients included in this study a physician-based diagnosis of CSU? How was this verified, especially for those patients recruited via the patient support group?

-          The most common comorbidity of a CSU is a CINDU -  Have patients with concomitant CINDU been excluded? Concomitant CINDU, especially cholinergic urticaria, heat urticaria, or symptomatic dermographism could additionally affect sleep quality, e.g., sleeping in bed with a partner, night sweats, etc. Concomitant CINDU also has an additional strong impact on sexual impairment.

-          The marital status in Table 1 distinguishes between being a couple and being single. "with couple" is not a correct designation; furthermore, marital status is not necessarily decisive for whether a bed is shared with someone else or not. It should be better specified what the authors had in mind by applying this way of stratifying patients.

-          Line 115: The Numeric Rating Scale is usually used from 0 to 10 not 1 to 10.

-          It is desirable to indicate a correlation coefficient in the Figures.

-          The DLQI is used in the figures. Why not the CU-QoL, which is specifically validated for patients with chronic urticaria?

-          Table 2 lacks an explanation of the values that are separated from each other by a slash in parentheses.

-          „Occupation vs. Employed“ it is not clear whether exactly these terms refer to when mentioned as they are mentioned in the Table.  

Author Response

Dear reviewer, 

The authors of the manuscript would like to thank you for your comments, as they allow us to improve the scientific quality of our research. Below, you can see a point-by-point response. 

  • The reference of CUQOL has been properly included.
  • The mistake of "AA" has been corrected.
  • The term "emotional status disorder" has been modified. 
  • The percentage of patients has been changed to the number of patients in the abstract.
  • The long UCT was collected. As the analysis performed is not dichotomic, but continuous the cut-off point is not of interest in this case. 
  • The language has been corrected by a native speaker.
  • A reference has been added with regard to the EAACI/EDF guidelines for the management of urticaria, 2022.
  • 10% of the patients chose not to take any drug for CSU when the survey was performed. This personal decision was variable in each patient (fear of treatments, tired of trying treatments without improvement, feeling better despite not having total control of the disease). With respect to disease severity and quality of life, there are both controlled and uncontrolled cases within this group of patients, although these data have not been described in the manuscript because they are not considered to be of interest.

  • "Protocolized follow-up" means that the patients were offered to participate after the dermatologists consultation for those who were collected via Dermatology consultations. Those patients recruited via patient support group were confirmed to be diagnosed as CSU by asking them for a official diagnosis of CSU made by a doctor. This paragraph has been modified to avoid understanding mistakes.
  • Patients with concomitant CINDU were excluded for this study. There is an on-going study for assessing the quality of life of those patients with CINDU. 
  • Marital status, or any relationship status is very difficult to define whatever the set of criteria applied, since there are also relationships composed of more than two people, open relationships, long-distance relationships and a multitude of other types of relationships. In this case, as in all other published studies on the subject, a simplistic reduction of relationships is made, distinguishing only whether people consider themselves to be in an "active" relationship or not. It would be impossible to adequately define whether or not people sleep together, live together 100% of the time, have a partner in addition to their regular partner, etc. Therefore, it was decided to simply stratify the patients in a simple way as shown in the table .
  • NRS has been corrected (0-10).
  • We have changed from DLQI to CUQOL in the figures.
  • The explanation has been added in table 2.
  • An explanation has been added with regard to "occupation" in the table.